# The diagnosis and management of dehydration in children with wasting or nutritional edema: A systematic review

**Adino Tesfahun Tsegaye**[1]*, **Patricia B. Pavlinac**[2], **Judd L. Walson**[3], **Kirkby D. Tickell**[2]

**1** Department of Epidemiology, University of Washington, Seattle, Washington, United States of America,
**2** Departments of Global Health, University of Washington, Seattle, Washington, United States of America,
**3** Departments of Global Health, Medicine (Infectious Diseases), Pediatrics and Epidemiology, University of Washington, Seattle, Washington, United States of America

* adino@uw.edu

**Data Availability Statement:** All study data is included in the manuscript and supplemental files.

**Funding:** This systematic review was contracted by the World Health Organization (202710538), which

## Abstract

Dehydration is a major cause of death among children with wasting and diarrhea. We reviewed the evidence for the identification and management of dehydration among these children. Two systematic reviews were conducted to assess 1) the diagnostic performance of clinical signs or algorithms intended to measure dehydration, and 2) the efficacy and safety of low-osmolarity ORS versus ReSoMal on mortality, treatment failure, time to full rehydration, and electrolyte disturbances (management review). We searched PubMed/Medline, Embase, and Global Index Medicus for studies enrolling children 0–60 months old with wasting and diarrhea. The diagnostic review included four studies. Two studies found the Integrated Management of Childhood Illness (IMCI) and the Dehydration: Assessing Kids Accurately (DHAKA) algorithms had similar diagnostic performance, but both algorithms had high false positive rates for moderate (41% and 35%, respectively) and severe (76% and 82%, respectively) dehydration. One further IMCI algorithm study found a 23% false positive rate for moderate dehydration. The management review included six trials. One trial directly compared low osmolarity ORS to ReSoMal and found no difference in treatment failure rates, although ReSoMal had a shorter duration of treatment (16.1 vs. 19.6 hours, p = 0.036) and a higher incidence of hyponatremia. Both fluids failed to correct a substantial number of hypokalemia cases across studies. In conclusion, the IMCI dehydration assessment has comparable performance to other algorithms among wasted children. Low osmolarity ORS may be an alternative to ReSoMal for children with severe wasting, but might require additional potassium to combat hypokalemia.

## Introduction

One-third of children who die from diarrhea each year have moderate or severe wasting [1, 2]. Dehydration is an important pathway leading to diarrhea-associated mortality [3], and its identification and treatment remain the main focus of diarrhea management guidelines. The diagnosis of dehydration in children includes signs which are common among children with

was awarded to KDT and PBP. The funders had no role in study design, data collection and analysis, decision to publish, or preparation of the manuscript.

**Competing interests:** The authors have declared that no competing interests exist.

wasting, irrespective of their hydration status [4, 5], indicating children with severe wasting may be frequently misdiagnosed with dehydration. These children may also be more vulnerable to fluid overload than other children, suggesting that misclassified dehydration among severely wasted children may result in unnecessary adverse effects, including pulmonary edema and death [6]. Improving the accuracy of dehydration assessments in wasted children may result in improved outcomes in this vulnerable population.

Low-osmolarity oral rehydration solution (ORS) is the cornerstone of diarrhea management, but guidelines recommend that children with severe wasting and dehydration (without shock or suspected cholera) be given Rehydration Solution for Malnourished (ReSoMal). ReSoMal was designed to rehydrate children with severe wasting while minimizing the risk of fluid overload and was proven to be superior to old standard ORS containing 15 mmol/L more sodium than low-osmolarity ORS [4, 6, 7]. It is unclear if ReSoMal is superior to low-osmolarity ORS for rehydrating children with severe wasting [8].

In addition to children with severe wasting, those with moderate wasting are also at increased risk of mortality during episodes of diarrhea [6]. Recommendations for the management of moderate wasting default to giving the low-osmolarity ORS used for children without wasting. This systematic review summarizes the evidence for the identification and management of dehydration among children with either moderate or severe wasting.

## Methods

We conducted two systematic reviews (PROSPERO ID: CRD42021276133). The first examined the diagnosis of dehydration among children with diarrhea and wasting. The second compared low-osmolarity ORS to ReSoMal for rehydrating these children. Both reviews included studies enrolling children 0–60 months old with diarrhea and moderate or severe wasting diarrhea (Table 1).

We searched PubMed/Medline, Embase, and Global Index Medicus for abstracts, full-text articles, and pre-prints written in English, French, and Spanish. No date restriction was applied (Search terms: Tables A & B in S1 Text) and articles published until February 2023 were included. Registries were not searched, but the citations of identified papers were reviewed to ensure no critical literature had been missed.

Two authors (ATT, KDT) screened titles and abstracts for articles meeting the inclusion criteria. Disagreements were discussed, and when unresolved, a third author (PBP) held the decisive vote. Both reviews excluded studies that did not report relevant results for children under 60 months of age or if they did not present results for children with wasting. The management review excluded studies focused on intravenous rehydration, or cholera/profuse watery diarrhea and those that did not use at least one of the solutions of interest (low-osmolarity ORS to ReSoMal, Table 2). Trials comparing one of the fluids to a solution other than low-osmolarity ORS or ReSoMal were included to gain a fuller understanding of the treatment outcomes of children given these fluids.

### Diagnostic review

This review's primary outcomes were the sensitivity and specificity of an index test compared to an eligible reference standard. Index tests were any clinical sign or collection of signs intended to measure dehydration. Eligible reference standards were pre-post rehydration weight, electrolyte imbalance, metabolic acidosis, or any established diagnostic criteria (e.g., WHO Integrated Management of Childhood Illness (IMCI) algorithm). The area under the curve (AUC), true positive, true negative, false positive, and false negative results were

**Table 1. PICO criteria.**

| For the diagnosis of dehydration review | |
| --- | --- |
| Population | Children aged 0–59 months, with moderate or severe wasting or edema or growth faltering and dehydration |
| Intervention (Index test) | Individual clinical sign or collection of signs intended to measure dehydration. |
| Comparison (Reference standard) * | Pre-post rehydration weight, electrolyte imbalance, renal function tests, metabolic acidosis, or any established diagnostic criteria (for e.g., WHO IMCI algorithm). |
| Outcomes | Sensitivity/Specificity |
| | Positive predictive value/false positive rate |
| | Area under the curve |
| For the management of dehydration review | |
| Population | Children aged 0–59 months*, with moderate or severe wasting or edema and dehydration by current or previous WHO, but who were not shocked and excluding children with cholera or profuse watery diarrhea; analysis stratified by nutritional status |
| Intervention | Standard low osmolarity ORS |
| Control/Comparison | ReSoMal or diluted standard WHO low-osmolarity ORS |
| Outcomes | Mortality |
| | Clinicial deteriorations, defined by withdrawal of the assigned oral fluid for clinical reasons, including development of any danger sign (obstructed breathing, respiratory distress, cyanosis, shock, severe anemia, convulsion, severe dehydration, profuse watery diarrhea, vomiting, and/or impaired consciousness) |
| | Duration of diarrhea |
| | Time to full rehydration |
| | Morbidity or recovery from co-morbidity |
| | Duration of hospital stay or time to discharge |
| | Weight change |

*There is no widely implemented gold standard for the assessment of dehydration among wasted children

secondary outcomes. Observational, quasi-experimental, and randomized studies comparing the accuracy of one or more index tests were eligible for the diagnostic review.

## Oral fluid management

The management review's outcomes were mortality, treatment failure (withdrawal of assigned fluid), time to full rehydration, weight change, morbidity or recovery from co-morbidity, and electrolyte disturbances during rehydration (including hyponatremia, hypernatremia, hypokalemia, hyperkalemia). Numerator and denominator data, proportions, odds ratio (OR), relative risk (RR), or hazard ratio (HR) were retrieved (Table 2). Only randomized control trials were included.

**Data extraction and synthesis.** Data abstraction templates were designed prior to full text review (Table A in S1 Text). Risk of bias was assessed using the QUADAS-2 tool for the diagnostic review. The management review used the Risk of Bias scale (RoB2, Tables C1-C3 in S1 Text) [9]. Data and risk of bias abstractions were conducted by two reviewers (ATT & KDT), and discordance was resolved through discussion and the third reviewer. The certainty of evidence was assessed via the GRADE process [9]. Meta-analysis was not attempted as each review included too few studies reported on the same outcome.

**Subgroup analyses.** Subgroups of interest were children with moderate wasting, severe wasting, or oedema, and those under six months old.

**Table 2. Inclusion and exclusion criteria.**

| Inclusion criteria | Exclusion criteria |
|---|---|
| **For both reviews** | |
| Moderate or severe wasting or edema and dehydration, by current or previous WHO definitions | No raw numbers or effect estimates reported |
| Written in English, French, and Spanish | Written in other languages |
| Published any time | |
| **For the diagnosis of dehydration review only** | |
| Age 0–5 years | Studies do not include report findings among individuals in this age range |
| Studies that compare any clinical symptom/sign/test to named reference standard for dehydration | Does not use a relevant reference standard for dehydration. |
| Includes a dehydration reference standard of at least one of: reported difference in pre-post rehydration weight, electrolyte imbalance, renal function test, metabolic acidosis, or an established criterion | Study included fewer than 5 children meeting with diarrhea and wasting |
| Reported raw numbers, proportions, and/or sensitivity, specificity, positive predictive value comparing signs/symptoms/test to dehydration reference standard | |
| **For the management of dehydration review only** | |
| Age 6-months to 10 years | Studies do not include report findings among individuals in this age range |
| Reported proportions, OR, RR, or HR | Contains only data relevant to intravenous rehydration (e.g., shock), or cholera/profuse watery diarrhea. Or, it is impossible to disaggregate these data from the desired data. |
| Comparison of ReSoMal and WHO low-osmolarity ORS | Used neither of the solutions or only of the solutions without comparison to another product. |
| Reported mortality and/or adverse outcomes (including electrolyte disturbances) | Reported other outcomes excluding mortality or adverse outcomes (including electrolyte disturbances) |

*Papers that were done in a different population or reported a different outcome but have relevance to the study findings were archived and included in the formal discussion.

## Results

### Diagnostic review

Of the 2,903 articles identified, three full texts and one abstract met the inclusion criteria (Fig 1). The studies were from India (Nagpal [1992] & Nijhawan [2020]), Bangladesh (Skrable [2017]), and South Africa (Beatty [1974], Table 3). Wasting definitions and inclusion of kwashiorkor patients varied across the studies. Diagnostic review: Beatty (1974) used Boston weight-for-age percentile (<50 percentile), and Nagpal (1992) used Gomez's weight-for-age grades (grades III&IV), Skrable (2017) used mid-upper arm circumference (MUAC <12.5cm–wasted, <11.5cm–severely wasted), while Nijhawan (2020) did not define wasting. Skrable (2017) excluded children with nutritional oedema [10], while the other studies did not comment on the inclusion or prevalence of nutritional oedema.

Three of the studies evaluated existing algorithms (IMCI, DHAKA, or the Clinical Dehydration Scale (CDS)) [10–12], while Beatty (1974) defined moderate dehydration as any one of the following signs: loss of tissue turgor, sunken eyes, sunken fontanelle, or dry mucous membranes, and severe dehydration as more than one of the those signs or signs of peripheral vascular collapse or shock [13]. Skrable (2017), also evaluated individual clinical signs [10].

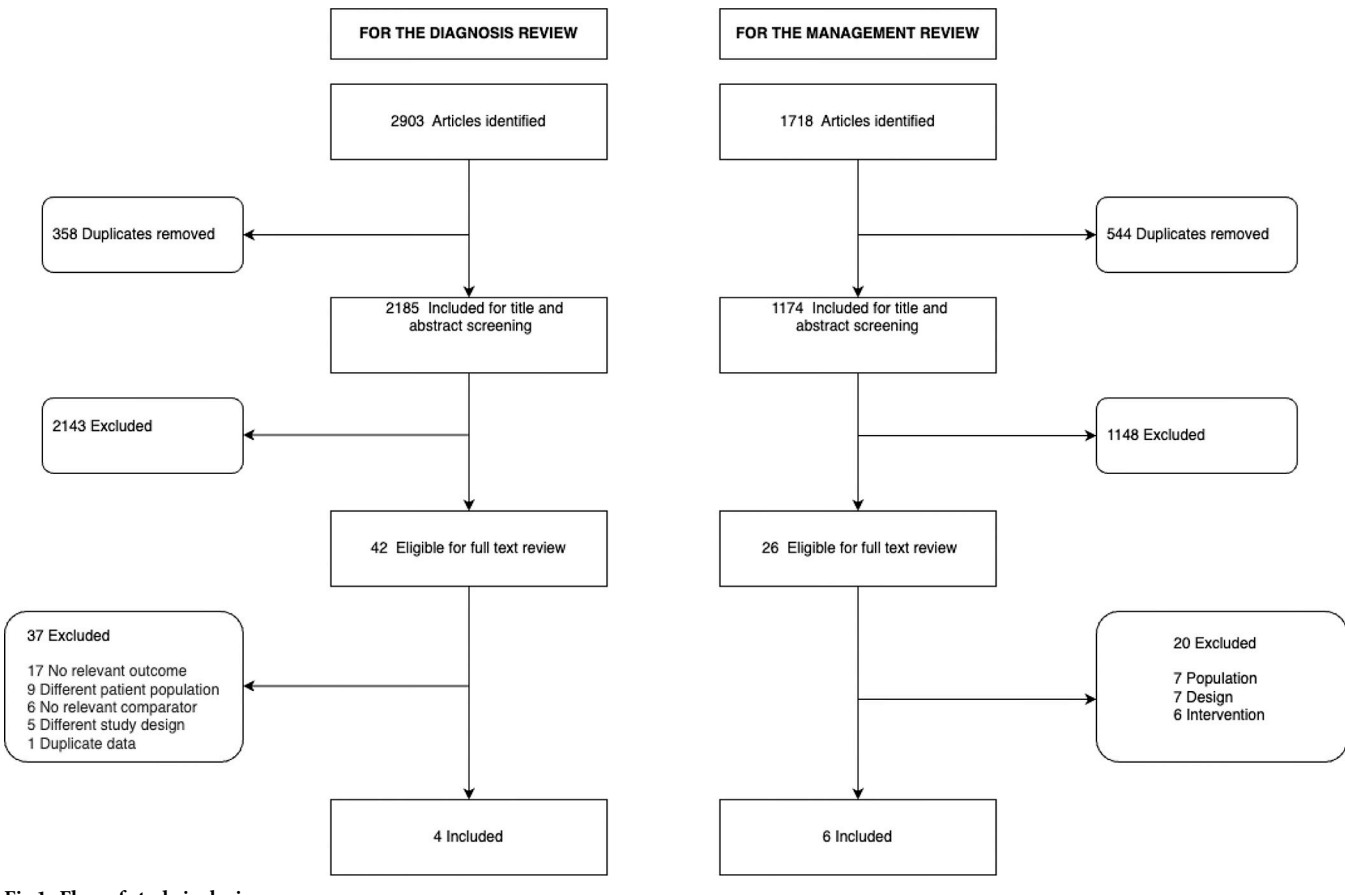

**Fig 1. Flow of study inclusion.**

The reference standard in Beatty (1974), Nagpal (1992), and Skrable (2017) was pre- and post-rehydration weight change. Only Skrable (2017) defined cut-offs for this reference test (no dehydration: weight change <3%, any dehydration weight change ≥3%, severe dehydration weight change ≥9%). Beatty (1974) & Nagpal (1992) described mean percentage change in weight within index test categories. Nijhawan (2020) used the WHO dehydration classification as the reference test.

**IMCI algorithm performance.** Skrable (2017) reported the AUC of the IMCI algorithm for predicting any dehydration (weight change ≥3%) to be 0.71 (95% confidence intervals [95%CI]: 0.66–0.77), with a sensitivity of 0.97 (95%CI: 0.94–1.00) and a specificity of 0.15 (95%CI: 0.08–0.22). The positive predictive value was 0.59 (95%CI: 0.53–0.66), indicating that 41% of children meeting the IMCI some dehydration criteria were false positives [10]. Skrable (2017) found the sensitivity of IMCI dehydration criteria for detecting severe weight gain to be 0.76 (95% CI: 0.59–0.93), with a specificity of 0.72 (95% CI: 0.66–0.78). The positive predicted value of IMCI severe dehydration was 0.24 (95% CI: 0.14–0.33), suggesting that 76% of children meeting these criteria were false positives.

Nagpal (1992) enrolled children with IMCI defined mild and moderate dehydration and assessed the mean difference in pre-post rehydration weights. Children with IMCI-defined mild dehydration had a mean weight gain of 3.5%, and those with moderate dehydration had a mean weight gain of 5.6%. However, seven (23.3%) of the 30 children with moderate dehydration were false positives, indicating a positive predictive value of 0.77 (95%CI: 0.58–0.91).

**Table 3. Summary of studies included in the diagnostic review.**

| Author (Year) | Study design | Country | Age | Dehydration diagnosis/ reference standard | Number of participants | Index test | Effect estimates (95% CI) |
|---|---|---|---|---|---|---|---|
| [a]Scrable (2017) | Cohort | Bangladesh | <60 months | No dehydration (< 3%weight change)<br>Any dehydration (> = 3% weight change)<br>Severe dehydration (> 9% weight change) | 1282 (19% malnutrition, 81% not mal.)<br>1042- No malnutrition<br>182 –MAM<br>58 –SAM<br>1274 (for diarrhea) | Sunken eyes | RR:1.397 (0.98–2.188) |
| | | | | | | General appearance | RR:2.167 (1.649–2.849) |
| | | | | | | Heart rate | RR:1.482 (1.182–1.859) |
| | | | | | | Mucous membranes | RR:1.667 (1.257–2.212) |
| | | | | | | Radial Pulse | RR:1.642 (1.337–2.017) |
| | | | | | | Respirations | RR:1.923 (1.554–2.382) |
| | | | | | | Skin pinch | RR:1.947 (1.443–2.627) |
| | | | | | | Tears | RR:2.127 (1.594–2.838) |
| | | | | | | Thirst | RR:2.955 (1.221–7.147) |
| | | | | | | DHAKA | AUC:0.783 (0.723–0.843) |
| | | | | | | | Sensitivity (some): 0.88 (0.83–0.94) |
| | | | | | | | Specificity (some): 0.40 (0.30–0.49) |
| | | | | | | | PPV (some): 0.65 (0.58, 0.72) |
| | | | | | | | NPV (some):0.72(0.61, 0.84) |
| | | | | | | | Sensitivity (severe): 0.88 (0.75–1.00) |
| | | | | | | | Specificity (severe): 0.55 (0.48–0.62) |
| | | | | | | | PPV (severe): 0.18 (0.12–0.25) |
| | | | | | | | NPV (severe): 0.98 (0.95, 1.00) |
| | | | | | | IMCI | AUC:0.713 (0.659–0.768) |
| | | | | | | | Sensitivity (some): 0.97 (0.94–1.00) |
| | | | | | | | Specificity (some): 0.15 (0.08–0.22) |
| | | | | | | | PPV (some):0.59 (0.53, 0.66) |
| | | | | | | | NPV (some):0.80 (0.62,0.98) |
| | | | | | | | Sensitivity (severe): 0.76 (0.83–0.94) |
| | | | | | | | Specificity (severe): 0.88 (0.66–0.78) |
| | | | | | | | PPV (severe): 0.24 (0.14–0.33) |
| | | | | | | | NPV (severe):0.96 (0.93,0.99) |
| | | | | | | CDS | AUC:0.774 (0.714–0.834) |
| [b]Beatty (1974) | RCT | South Africa | 6 weeks to 4 years | 5% (Moderate)- if there were signs of a loss of tissue turgor, sunken eyes, sunken fontanelle, or dry mucous membranes.<br>10% (Severe) dehydration- more than one of the above signs or if there were signs of peripheral vascular collapse or shock. | 80 | Some Dehydration | Mean weight gain 5.8% (4.6%-7.0%) |
| | | | | | | Severe Dehydration | Mean weight gain 8.0% (7.3%-8.7%) |
| | | | | | | Hypernatremia | 2/7 |
| | | | | | | Marked Acidosis | Mean PH 7.15 |
| | | | | | | Mild Acidosis | Mean PH 7.26 |

*(Continued)*

**Table 3.** (Continued)

| Author (Year) | Study design | Country | Age | Dehydration diagnosis/ reference standard | Number of participants | Index test | Effect estimates (95% CI) |
|---|---|---|---|---|---|---|---|
| [c]Nijhawan (2020) | Cohort | India | 2 months to 5 years | WHO criteria | 503 | DHAKA | AUC: 0.92 |
| | | | | | | | Sensitivity: 84% |
| | | | | | | | Specificity: 100% |
| [d]Nagpal (1992) | Cohort | India | 1m – 4Y | WHO criteria | 50 Grade III PEM– 28 Grade IV PEM– 22 | Mild dehydration | Mean weight gain 3.5% |
| | | | | | | Moderate dehydration | Mean weight gain 6.0%, 7 children gained < 5% |
| | | | | | | | Moderate dehydration PPV: 0.77 (0.58, 0.91) |

**Diagnosis and classification of malnutrition:** a: based on MUAC, <12.5cm malnutrition, and <11.5cm severe acute malnutrition; b: undernutrition—<50% Weight for age on "Boston" scale, or albumin post-rehydration; c: NA–definition not given; d: severe acute malnutrition–weight for age <60% (Graded as Grade III/IV based on the 1972 recommendation of the Indian Academy of Pediatric)

AUC: Area under the curve; CDS: Clinical dehydration scale; DHAKA: Dehydration assessing kids accurately (IIMCI: Integrated management of childhood illness; NPV: Negative predictive value; PPV: Positive predictive value; PEM: Protein energy malnutrition; RCT: Randomized controlled trial; RR: Relative risk.

Beatty (1974), whose clinical signs were similar to IMCI but with a different scoring system, found children initially classified as moderately dehydrated had a mean weight gain of 5.8% (95%CI: 4.6%-7.0%) after rehydration, and those categorized as severely dehydrated had a mean weight gain of 8.0% (95%CI: 7.3%-8.7%). Beatty (1974) did not report the false positive rate or related results.

**DHAKA score performance.** In Skrable (2017), the AUC of the DHAKA score for predicting any dehydration (weight change ≥3%) was 0.78 (95%CI: 0.72–0.84), the sensitivity was 0.88 (95%CI: 0.83–0.94), and the specificity was 0.40 (95%CI: 0.30–0.49). The positive predictive value was 0.65 (95%CI: 0.58–0.72), indicating that 35% of the children meeting the DHAKA some dehydration criteria were false positives. For severe dehydration, the DHAKA score had a sensitivity of 0.88 (95%CI: 0.75–1.00) and a specificity of 0.55 (95% CI: 0.48–0.62). The positive predictive value for severe dehydration was 0.18 (95% CI: 0.12–0.25), indicating that 82% of children with DHAKA-defined severe dehydration were false positives [10].

In Nijhawan (2020), the DHAKA score had a sensitivity of 84%, a specificity of 100%, and an AUC of 0.92 (95% CI: 0.81–0.98) compared to the WHO algorithm.

**CDS scale.** Skrable (2017) reported the AUC of the CDS scale as 0.77 (95% CI: 0.71–0.83). No cutoffs for the CDS were available to calculate the sensitivity, specificity, or positive predicted value.

**Individual clinical sign performance.** Skrable (2017) evaluated individual signs of dehydration. All clinical signs, with the exception of sunken eyes, had a significant association with the presence of a ≥3% weight gain during rehydration (Table D in S1 Text).

**Subgroups.** One study (Skrable [2017]) reported results by subgroups noting that their results did not change when they stratified by severe (MUAC <11.5cm, N = 58) and moderate wasting (MUAC <12.5cm, N = 182), and that the AUCs of the IMCI, DHAKA, CDS were not different between children with and without wasting.

**Certainty of evidence.** Details of the certainty of evidence assessment are given in Tables E1 to E4 in S1 Text, but all evidence in the diagnostic review was judged to have very low certainty of evidence (Table C1 in S1 Text).

## Oral fluid management review

Among 1,718 articles (Fig 1, Table F in S1 Text), only six studies—two from India, two from Bangladesh, one from Mexico, and one from Venezuela (Table 4) met the inclusion criteria [14, 15]. Definitions of wasting and the inclusion of kwashiorkor varied across the studies. Two studies defined wasting by the WHO criteria (Kumar 2015, Conde 2005) [14, 16], while the remaining studies used weight-for-length or weight-for-age z-scores <-3 (Alam 2015) [17], a Gomez criteria grade II or III (Faure 1990), a weight-for-length < 70% of the median (Alam 2003) [18], and a weight-for-length <60% of the Harvard/Boston scale median (Dutta 2001) [19]. Faure (1990) was the only trial to include children with moderate wasting [15]. Prevalence of kwashiorkor varied across the studies with one study excluding these children, two studies not commenting on kwashiorkor, and the remaining studies reporting a prevalence between 22% and 73%. Studies included infants and children aged 1 to 48 months.

Kumar (2015) was the only study to compare low-osmolarity ORS to ReSoMal directly [16]. The low osmolarity ORS in this trial had 20mmol/L more potassium than the current WHO formulation. Five studies included one of the two interventions of interest without comparison to the other solution. Two studies (Alam 2003, Conde 2005) included ReSoMal arms, both utilizing the WHO-recommended formulation [14, 18]. Three studies used a low-osmolarity ORS, but with formulations that deviated from the current WHO recommendations (Table G in S1 Text).

**Treatment efficacy: Four studied assessed treatment efficacy.** The trial comparing ReSoMal to low-osmolarity ORS (Kumar, 2015) [16] found no difference in *treatment failure* rates, with both arms having 3/55 (5%) children requiring intravenous fluid. One additional study, including a ReSoMal arm without comparison to low-osmolarity ORS, found 7/65 (11%) failed treatment. Three studies of low osmolarity ORS without comparison to ReSoMal reported treatment failure among 0/32 (0%), 1/82 (1%), and 3/63 (5%) children.

In Kumar (2015), ReSoMal was found to have a significantly shorter *duration of treatment* (16.1 hours to 19.6 hours, p = 0.036) compared to low-osmolarity ORS. No studies reported on *duration of diarrhea*; however, Kumar (2015) noted both arms had similar median stool frequencies [16]. Other outcomes of interest (mortality, morbidity/recovery from co-morbidity, weight change) were not reported in the included studies.

## Adverse effects

**Fluid overload.** One study including ReSoMal without a comparison low-osmolarity ORS found 3/65 (5%) children treated with ReSoMal became overloaded.

**Hyponatremia.** Five studied reported about serum sodium level. The trial directly comparing the solutions found ReSoMal to have a higher incidence of hyponatremia (15.4% vs. 1.9%, p = 0.03) than low-osmolarity ORS. However, no hyponatremia cases were severe/symptomatic (<130mmol/L). Notably, children with cholera or high purge rates were excluded from this trial.

In one study utilizing ReSoMal without comparison to low-osmolarity ORS, 24/65 (37%) children receiving ReSoMal were hyponatremic after 24 hours of treatment and 17/65 (26%) after 48 hours, with three (18%) children developing severe hyponatremia and one additional child having a hyponatremia-attributed seizure. All severe hyponatremia cases in this study were among children with cholera or high purge rates. The second study of ReSoMal, without comparison to low-osmolarity ORS, found mean sodiums of 131.4 mmol/L, 130.9 mmol/L, and 131.8 mmol/L at treatment initiation, 4 and 8 hours, respectively, but did not note if any children became severely hyponatremic.

Two of the three studies using low osmolarity ORS reported serum sodium levels during rehydration. One study found the post-rehydration mean sodium to be 134.4 mmol/L. The

**Table 4. Summary of studies included in the management review.**

| Author (Year) | Study design | Country | Population | Intervention | N | Outcomes | Low Osmolarity ORS | ReSoMal | P-value |
|---|---|---|---|---|---|---|---|---|---|
| **Direct Comparison** | | | | | | | | | |
| Kumar (2015) [16] | Single center, open-label RCT | India | Age: 6-59 months, severely wasted (WHO definition: WHZ <-3, MUAC 11.5cm, and/or oedema), with diarrhea | ReSoMal<br><br>Low Osmolarity ORS (Na: 75mmol, K: 40mmol/l[1], Osmo: 265 mmol/l)<br><br>Standard care was administered in addition to the above interventions. | 110 (55/arm) | Clinical deteriorations[2] | 3/55 (5%) | 3/55 (5%) | Not reported |
| | | | | | | Time to rehydration[3] | 19.6hrs | 16.1hrs | 0.036 |
| | | | | | | Hyponatraemia[4] | 1/52 (1.9%) | 8/52 (15.4%) | 0.03 |
| | | | | | | Hypernatraemia[4] | 1/52 (1.9%) | 0/52 (0%) | Not reported |
| | | | | | | Hypokalaemia[4] | 5/52 (9.6%) | 9/52 (17.3%) | 0.25 |
| | | | | | | Mortality | 0/52 (0%) | 0/52 (0%) | Not reported |
| Faure (1990) [15] | RCT | Mexico | 1-36 months | Low Osmolarity ORS (Na: 60mmo/l l, K: 25mmol/l, Osmo: 240 mmol/l) | 82 | Treatment failure | 1 (1.2%) | | -- |
| | | | | | | Time to rehydration (Gomez II) | 5.33 hrs +/- 1.75 | | |
| | | | | | | Time to rehydration (Gomez III) | 4.33 hrs +/- 0.57 | | |
| | | | | | | Post rehydration sodium (Gomez II) | 130.6 +/- 5.1 | | -- |
| | | | | | | Post rehydration sodium (Gomez III) | 140.9 +/- 5.5 | | -- |
| | | | | | | Post rehydration potassium (Gomez II) | 4.22 +/- 1.06 | | -- |
| | | | | | | Post rehydration potassium (Gomez III) | 4.60 +/- 1.11 | | -- |
| | | | | | | Correction of potassium (Grade II) | 2 of 4 cases | | -- |
| Dutta (2001) [19] | RCT | India | 6-48 months | Low Osmolarity ORS (Na: 60mmo/l l, K: 20mmol/l, Osmo: 224 mmol/l) | 32 | Recovery in five days | 32 (100%) | | -- |
| | | | | | | Median time to recovery | 36 hrs | | |
| | | | | | | Duration of diarrhea | 41.5 hrs | | -- |
| | | | | | | Fluid intake | 61.2 ml/kg/day | | |
| | | | | | | Weight gain (%) | 4.30% | | -- |
| | | | | | | Pre-treatment sodium | 130.0 mmol/l +/- 3.3 | | |
| | | | | | | Post rehydration sodium | 134.4 mmol/l +/- 3.1 | | -- |
| | | | | | | Pre-treatment potassium | 3.1 mmol/l +/- 0.3 | | |
| | | | | | | Post rehydration potassium | 3.5 mmol/l +/- 0.3 | | -- |
| Alam (2003) [18] | RCT | Bangladesh | 6-36 months | ReSoMal (Na: 45mmol, K: 40mmol/l, Osmo: 300 mmol/l) | 65 | Treatment failure | 7 (11%) | | -- |
| | | | | | | Fluid overload | 3 (5%) | | -- |
| | | | | | | Hyponatraemia at 24hrs | 24/62 (39%) | | -- |
| | | | | | | Hyponatraemia at 48hrs | 17/62 (29%) | | -- |
| | | | | | | Severe hyponatraemia | 3 (5%) | | -- |
| | | | | | | Hyponatraemia non-cholera @ 24hour | 7 of 47 | | |
| | | | | | | Hyponatraemia non-cholera @ 48hour | 4 of 47 | | -- |
| | | | | | | Hypokalemia corrected at 24hrs | 14 of 38 | | -- |
| | | | | | | Hypokalemia corrected at 48 hrs | 18 of 38 | | |
| Conde (2005) [14] | RCT | Venezuela | 2-23 months | ReSoMal (Na: 45mmol, K: 40mmol/l, Osmo: 300 mmol/l)3 | 15 | Mean serum sodium | 131.4 mmol/l (baseline) 130.9 mmol/l (4hrs) 131.8 mmol/l 8 (hrs) | | -- -- -- |
| | | | | | | Mean serum potassium | 2.8 mmol/l (baseline) 3.4 mmol/l (4hrs) 3.8 mmol/l (8 hrs) | | -- -- -- |
| Alam (2015) [17] | RCT | Bangladesh | 6-36 mnths | Low Osmolarity ORS (Na: 75mmo/l l, K: 40mmol/l, Osmo: 647 mmol/l)[4] | 63 | Treatment failure | 3 (5%) | | -- |
| | | | | | | Severe hypokalemia | 3 (5%) | | |

[1]In studies with only one relevant arm, only the N for that arm is presented. [2]Only p-values for comparison between ReSoMaL and low osmolarity ORS are given. [3]Conde (2005) [14] does not list the specific formulation of ReSoMal but indicates it was made according to WHO recommendations. [4]Alam (2015) include 300mmol/L of zinc and

[1]20mmol/l added to standard low osmolarity ORS regimen. [2]Clinical deteriorations: Two children in the low-osmolarity ORS group became shocked and one developed hypernatraemia. All three clinical deteriorations in the ReSoMal were children who developed shock. These clinical deteriorations were described as treatment failures in the manuscript. [3]Three or more signs of improved hydration: tears, no longer thirsty, slowed heart rate, slowed respiratory rate, less lethargic, skin pinch less slow. [4]Values used to define electrolyte discrepancies were not explicitly defined in the manuscript.

NB: Faure (1990) [15] used a lower sodium concentration (60mmol/L) and slightly higher potassium concentration (25mmol/L). Dutta (2001) [19] also used a lower sodium concentration (60mmol/L). Finally, Alam (2015) [15], added zinc and copper, and slightly modified the chloride, citrate and glucose content of a low-osmolarity ORS. No studies reported data stratified by the pre-specified subgroups of age, moderate wasting or kwashiorkor.

second study stratified results by Gomez criteria and found mean sodium of 130.6 (+/- 5.1) mmol/L among with grade II and 140.9 (+/- 5.5) mmol/L among grade III wasting.

**Hypernatremia.** The trial comparing the solutions noted 1/55 (2%) children given low osmolarity ORS, and none of the ReSoMal arm, became hypernatremic. No cases of hypernatremia were noted in the other low osmolarity ORS studies, and the two studies reporting post-rehydration mean serum sodium for this intervention were 134.4 mmol/L and 140.9 +/- 5.50.

**Hypokalemia.** Six studied reported about participants potassium level post-rehydration. In the trial comparing the two products, both solutions contained 40 mmol/L of potassium, and there was no significant difference in observed hypokalemia between the arms. The rates of hypokalemia were high in both arms, with nine (17.3%) ReSoMal and five (9.6%) low osmolarity ORS found to be hypokalemia after 48 hours of treatment.

One of two studies including ReSoMal without comparison to low osmolarity ORS, found the mean potassium to steadily increase during rehydration, from 2.8 at baseline to 3.8 at 8 hours of treatment. However, the second study observed that ReSoMal only corrected 14/38 (37%) hypokalemia cases after 24 hours of treatment, and one child was withdrawn for developing symptomatic hypokalemia. Despite these deficits, the mean serum potassium in their study rose from 3.3 (+/-1) to 4.0 (+/-1) mmol/L during rehydration.

The two studies using low osmolarity ORS without comparison to ReSoMal reported post-treatment mean potassium to be within the normal range. One study noted that 2/4 (50%) cases of hypokalemia were resolved during rehydration. The second study noted that 3/63 (5%) children were withdrawn from low osmolarity ORS due to severe hypokalemia.

**Certainty of evidence.** The management review outcomes were found to be of low or very low certainty evidence (Table C2 in S1 Text, full description Table H in S1 Text).

## Discussion

This review found the currently recommended WHO/IMCI dehydration algorithm to have comparable performance to available alternatives, such as DHAKA and CDS, among children with wasting (algorithms defined in S1 Text). All algorithms had high false positivity rates, which could lead to the unnecessary administration of fluids in a population at potential risk of iatrogenic adverse events, including fluid overload and electrolyte disturbances. Although recent cardiac studies have challenged the idea that wasted children are prone to fluid overload [20, 21], they remain vulnerable to changes in their electrolyte status during rehydration. The included studies identified multiple cases of severe hyponatremia, hypernatremia, and hypokalemia. The heterogeneity of electrolyte disturbances, the high false positive rate of dehydration algorithms, and the potential for adverse events among children with wasting should encourage healthcare providers to remain vigilant during rehydration and use timely electrolyte testing where possible.

WHO currently recommends the use of ReSoMal for children with severe acute malnutrition instead of low osmolarity ORS which is used for all other children. The management review did not find substantial differences in key treatment outcomes between low osmolarity ORS and ReSoMal among wasted children with diarrhea. ReSoMal may be associated with a shorter duration of treatment but potentially poses a higher risk of mild hyponatremia [22]. Another recent systematic review raised concerns about the frequency of hyponatremia among children treated with ReSoMal [8]. We identified similar concerns and included two additional studies that also found evidence of hyponatremia among children treated with ReSoMal. There does appear to be a high rate of mild hyponatremia when using ReSoMal, but its clinical importance is unclear. Severe cases appear to have only occurred among children with cholera or very high purge rates, which are already contraindications to ReSoMal in WHO guidance [6].

The Indian Academy of Pediatrics [23] and the International Centre for Diarrhoeal Disease Research, Bangladesh [24], already use low osmolarity ORS as their default treatment for children with wasting and dehydration. ReSoMal is currently three times more expensive than low osmolarity ORS [25], and is particularly vulnerable to stockouts. A recent Zimbabwean study found 80% of provinces did not have ReSoMal, while in Rwanda, a survey found five of eight acute care facilities had no ReSoMal [26, 27]. During ReSoMal stockouts, WHO guidelines recommend that low osmolarity ORS be dissolved in an additional liter of water, and more potassium be added to make half-strength low osmolarity ORS. Across our review, the Houston *et al*. systematic review, and the 2013 severe acute malnutrition guideline update, no trials testing the use of half-strength low osmolarity ORS have been identified [6, 8]. Low osmolarity ORS may be a cheaper, widely available alternative to ReSoMal and half-strength low osmolarity ORS.

Low osmolarity ORS contains 20 mmol/L less potassium than ReSoMal. In three of four studies using low osmolarity ORS, supplemental potassium was added to the solution. Despite these modifications, severe and symptomatic hypokalemia were common in both solutions, as was persistent asymptomatic hypokalemia. Given that solutions containing 40mmol/L of potassium have demonstrated borderline effectiveness at correcting hypokalemia, we raise concerns that the concentration of potassium in WHO-formulation low osmolarity ORS (20 mmol/L) may be insufficient. This may suggest that 20 mmol/L of potassium should be added to low osmolarity ORS when used among children with wasting.

## Strength and limitations

This review had several strengths, including a broad search strategy, but there were also several limitations. Data directly addressing children with moderate wasting, kwashiorkor, or those under six months of age were insufficient to comment on these groups meaningfully. We were not able to pool results across studies, which prevented us from testing for publication bias. For the diagnosis review, we identified a limited number of studies, and the manuscripts were often not compliant with the STARD guidelines [28]. Pre- and post-rehydration weights may introduce misclassification due to incomplete rehydration, food intake, passing stool, or urinary voiding. For the management review, we only included randomized controlled trials but did accept single arms from trials to provide indirect evidence of efficacy. Comparing arms across trials does not provide high-quality evidence due to differences between populations and clinical practices across studies.

## Conclusion

The current evidence suggests that the WHO/IMCI dehydration assessment for children with wasting has comparable performance to published alternatives. However, it results in a substantial number of false positive diagnoses, which warrant continued caution and close monitoring during rehydration. There is little evidence to suggest clinically important differences in outcomes between low osmolarity ORS and ReSoMal, although there was insufficient evidence to demonstrate true non-inferiority. ReSoMal may be associated with mild hyponatremia and should not be used in children with high purge rates. Low osmolarity ORS may be a viable alternative to ReSoMal for children with severe wasting, although the risk of hypokalemia due to the low potassium concentration of low osmolarity ORS is an important consideration. In many settings, low-osmolarity ORS may replace ReSoMal because it is cheaper and more available, and their efficacies appear similar.

## Supporting information

**S1 Checklist. Completed PRISMA checklist.**
(DOCX)

**S1 Text. Supplementary file.**
(DOCX)

## Acknowledgments

We thank the University of Washington Health sciences librarians for their technical support in developing the search terms. We also thank WHO staff Drs. Allison Daniel, Kirrily de Polnay, and Jaden Bandabenda for their technical support through the review process.

## Author Contributions

**Conceptualization:** Adino Tesfahun Tsegaye, Patricia B. Pavlinac, Judd L. Walson, Kirkby D. Tickell.

**Data curation:** Adino Tesfahun Tsegaye, Patricia B. Pavlinac, Kirkby D. Tickell.

**Formal analysis:** Adino Tesfahun Tsegaye, Patricia B. Pavlinac, Kirkby D. Tickell.

**Funding acquisition:** Adino Tesfahun Tsegaye, Patricia B. Pavlinac, Judd L. Walson, Kirkby D. Tickell.

**Investigation:** Adino Tesfahun Tsegaye, Patricia B. Pavlinac, Judd L. Walson, Kirkby D. Tickell.

**Methodology:** Adino Tesfahun Tsegaye, Patricia B. Pavlinac, Judd L. Walson, Kirkby D. Tickell.

**Project administration:** Adino Tesfahun Tsegaye, Patricia B. Pavlinac, Judd L. Walson, Kirkby D. Tickell.

**Resources:** Adino Tesfahun Tsegaye, Patricia B. Pavlinac, Judd L. Walson, Kirkby D. Tickell.

**Software:** Adino Tesfahun Tsegaye, Patricia B. Pavlinac, Kirkby D. Tickell.

**Supervision:** Adino Tesfahun Tsegaye, Patricia B. Pavlinac, Judd L. Walson, Kirkby D. Tickell.

**Validation:** Adino Tesfahun Tsegaye, Patricia B. Pavlinac, Judd L. Walson, Kirkby D. Tickell.

**Visualization:** Adino Tesfahun Tsegaye, Kirkby D. Tickell.

**Writing – original draft:** Adino Tesfahun Tsegaye, Kirkby D. Tickell.

**Writing – review & editing:** Adino Tesfahun Tsegaye, Patricia B. Pavlinac, Judd L. Walson, Kirkby D. Tickell.

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
