## [Decision Letter · Decision Letter 0]

26 May 2023

PGPH-D-23-00354

The Diagnosis and Management of Dehydration in Children with Wasting or Nutritional Edema: A Systematic Review

Dear Dr. Tsegaye,

Thank you for submitting your manuscript to PLOS Global Public Health. After careful consideration, we feel that it has merit but does not fully meet PLOS Global Public Health’s publication criteria as it currently stands. Therefore, we invite you to submit a revised version of the manuscript that addresses the points raised during the review process.

Please note that we have only been able to secure a single reviewer to assess your manuscript. We are issuing a decision on your manuscript at this point to prevent further delays in the evaluation of your manuscript. Please be aware that the editor who handles your revised manuscript might find it necessary to invite additional reviewers to assess this work once the revised manuscript is submitted. However, we will aim to proceed on the basis of this single review if possible. 

We look forward to receiving your revised manuscript.

Kind regards,

Steve Zimmerman, PhD

PLOS Staff Editor

Journal Requirements:

a. Please clarify all sources of funding (financial or material support) for your study. List the grants (with grant number) or organizations (with url) that supported your study, including funding received from your institution. 

b. State the initials, alongside each funding source, of each author to receive each grant.

c. State what role the funders took in the study. If the funders had no role in your study, please state: “The funders had no role in study design, data collection and analysis, decision to publish, or preparation of the manuscript.”

d. If any authors received a salary from any of your funders, please state which authors and which funders.

2. Please provide separate figure files in .tif or .eps format.

Additional Editor Comments (if provided):

Reviewers' comments:

Reviewer's Responses to Questions

**Comments to the Author**

1. Does this manuscript meet PLOS Global Public Health’s publication criteria? Is the manuscript technically sound, and do the data support the conclusions? The manuscript must describe methodologically and ethically rigorous research with conclusions that are appropriately drawn based on the data presented.

Reviewer #1: Yes

2. Has the statistical analysis been performed appropriately and rigorously?

Reviewer #1: N/A

3. Have the authors made all data underlying the findings in their manuscript fully available (please refer to the Data Availability Statement at the start of the manuscript PDF file)?

Reviewer #1: Yes

4. Is the manuscript presented in an intelligible fashion and written in standard English?

Reviewer #1: Yes

5. Review Comments to the Author

Reviewer #1: Respected authors,

Thank you very much for submitting your work at PLOS Global Public Health. I have thoroughly reviewed your work, and honestly I enjoyed reading your work. Overall, the concept of your work is exceptional, but it need some improvement in presentation.

I am expecting you will work on all the suggestions and will improve the quality and presentation as much as you can.

Abstract: No comments.

Introduction: No comments.

Methodology: In your methodology section, I found a lot of tables. I suggest you write your methodology according to the PRISMA guidelines for systematic/scoping review. You can visit following links:

1) http://www.prisma-statement.org/documents/PRISMA_2020_checklist.pdf

2) http://www.prisma-statement.org/Extensions/ScopingReviews

I hope this will aid you to refine your methodology.

Moreover, in your methodology do not forget to define your variables of interest.

Results:

In the result section, start a paragraph for study characteristics. If possible create a diagram, which describe how many studies assessed wasting and how many treatment interventions. Study location, study design, study setting, duration, etc.

Moreover, after presenting study characteristics for each review, I suggest you try to present each study measure and outcome and present its component in separate tables. Such as while presenting electrolytes level of children with moderate to severe dehydration, discuss the finding according to table.

Do not copy and paste same thing, rather than write hyponatrimia, hypernatremia, hypokalemia, hyperkalemia, and so on.

In this way, it would be easier to understand, of six studies how many studies examine sodium and potassium level, if X studies assessed sodium and potassium level then how many reported electrolytes imbalance and how many showed normal electrolytes. Same thing you need to do for other variables, such as fluid overload and treatment failure.

Discussion:

I feel that while writing discussion you need to elaborate further about the diagnostic signs and also about the managment.

Write strengths and limitation in a separate heading.

Add a heading of study implication.

Looking forward to see next version.

Best regards,

6. PLOS authors have the option to publish the peer review history of their article (what does this mean?). If published, this will include your full peer review and any attached files.

**Do you want your identity to be public for this peer review?** For information about this choice, including consent withdrawal, please see our Privacy Policy.

Reviewer #1: **Yes: **Asif Khaliq

---

## [Decision Letter · Decision Letter 1]

4 Sep 2023

PGPH-D-23-00354R1

The Diagnosis and Management of Dehydration in Children with Wasting or Nutritional Edema: A Systematic Review

Dear Dr. Tsegaye,

Thank you for submitting your manuscript to PLOS Global Public Health. After careful consideration, we feel that it has merit but does not fully meet PLOS Global Public Health’s publication criteria as it currently stands. Therefore, we invite you to submit a revised version of the manuscript that addresses the points raised during the review process.

We look forward to receiving your revised manuscript.

Kind regards,

Tinku Thomas, Ph.D

Academic Editor

Journal Requirements:

Additional Editor Comments (if provided):

Reviewers' comments:

Reviewer's Responses to Questions

**Comments to the Author**

1. If the authors have adequately addressed your comments raised in a previous round of review and you feel that this manuscript is now acceptable for publication, you may indicate that here to bypass the “Comments to the Author” section, enter your conflict of interest statement in the “Confidential to Editor” section, and submit your "Accept" recommendation.

Reviewer #2: All comments have been addressed

2. Does this manuscript meet PLOS Global Public Health’s publication criteria? Is the manuscript technically sound, and do the data support the conclusions? The manuscript must describe methodologically and ethically rigorous research with conclusions that are appropriately drawn based on the data presented.

Reviewer #2: Yes

3. Has the statistical analysis been performed appropriately and rigorously?

Reviewer #2: I don't know

4. Have the authors made all data underlying the findings in their manuscript fully available (please refer to the Data Availability Statement at the start of the manuscript PDF file)?

Reviewer #2: Yes

5. Is the manuscript presented in an intelligible fashion and written in standard English?

Reviewer #2: Yes

6. Review Comments to the Author

Reviewer #2: The article is well written

Please include the statistical methods in the article.

Results

How did the authors overcome bias

Can the results be represented in forest plot

Why only RCTs were included

7. PLOS authors have the option to publish the peer review history of their article (what does this mean?). If published, this will include your full peer review and any attached files.

**Do you want your identity to be public for this peer review?** For information about this choice, including consent withdrawal, please see our Privacy Policy.

Reviewer #2: No

---

## [Editor Report · Decision Letter 2]

4 Oct 2023

The Diagnosis and Management of Dehydration in Children with Wasting or Nutritional Edema: A Systematic Review

PGPH-D-23-00354R2

Dear Tsegaye,

We are pleased to inform you that your manuscript 'The Diagnosis and Management of Dehydration in Children with Wasting or Nutritional Edema: A Systematic Review' has been provisionally accepted for publication in PLOS Global Public Health.

Best regards,

Tinku Thomas, Ph.D

Academic Editor

I thank the authors for sufficiently addressing all the comments raised by the reviewers.